# Identification of the Role of TGR5 in the Regulation of Leydig Cell Homeostasis

**DOI:** 10.3390/ijms232315398

**Published:** 2022-12-06

**Authors:** Hélène Holota, Angélique De Haze, Emmanuelle Martinot, Melusine Monrose, Jean-Paul Saru, Françoise Caira, Claude Beaudoin, David H. Volle

**Affiliations:** INSERM U1103, CNRS UMR-6293, Université Clermont Auvergne, GReD Institute, Team-Volle, F-63001 Clermont-Ferrand, France

**Keywords:** bile acids, TGR5, testosterone, cholesterol esters

## Abstract

Understanding the regulation of the testicular endocrine function leading to testosterone production is a major objective as the alteration of endocrine function is associated with the development of many diseases such as infertility. In the last decades, it has been demonstrated that several endogenous molecules regulate the steroidogenic pathway. Among them, bile acids have recently emerged as local regulators of testicular physiology and particularly endocrine function. Bile acids act through the nuclear receptor FXRα (Farnesoid-X-receptor alpha; NR1H4) and the G-protein-coupled bile acid receptor (GPBAR-1; TGR5). While FXRα has been demonstrated to regulate testosterone synthesis within Leydig cells, no data are available regarding TGR5. Here, we investigated the potential role of TGR5 within Leydig cells using cell culture approaches combined with pharmacological exposure to the TGR5 agonist INT-777. The data show that activation of TGR5 results in a decrease in testosterone levels. TGR5 acts through the PKA pathway to regulate steroidogenesis. In addition, our data show that TGR5 activation leads to an increase in cholesterol ester levels. This suggests that altered lipid homeostasis may be a mechanism explaining the TGR5-induced decrease in testosterone levels. In conclusion, the present work highlights the impact of the TGR5 signaling pathway on testosterone production and reinforces the links between bile acid signaling pathways and the testicular endocrine function. The testicular bile acid pathways need to be further explored to increase our knowledge of pathologies associated with impaired testicular endocrine function, such as fertility disorders.

## 1. Introduction

One of the main roles of the testis is to produce sex steroids, corresponding to the endocrine function. These hormones play a major role in the control of testicular physiology but also in the regulation of many other physiological functions. Steroid synthesis is initiated from cholesterol. The latter is taken over by the transporters StAR (steroidogenic acute regulatory protein) or PBR (peripheral benzodiazepine receptor or translocator protein, TSPO); this allows the conversion of cholesterol to pregnenolone within the mitochondria [1,2]. Then, steroidogenesis proceeds through a cascade of enzymatic steps involving CYP11A1 (cytochrome P450, family 11, subfamily A, polypeptide 1), HSD3β (3β-hydroxysteroid dehydrogenase), and CYP17A1 (cytochrome P450, family 17, subfamily A, polypeptide 1) [3]. These different successive steps lead to the production of testosterone.

Bile acid homeostasis and dependent signaling pathways have emerged in recent years as key modulators of testicular physiology, with a major impact on the Leydig cell function [4]. Indeed, the nuclear receptor Farnesoid-X-receptor-α (FXRα; NR1H4) has been defined as a regulator of testicular endocrine function. FXRα activation leads to the repression of testicular steroidogenesis [5]. This negative impact of FXRα on steroid synthesis is mainly due to the regulation of *Shp (small heterodimer partner; NR0B2)* [6] and *Dax-1* (dosage-sensitive sex reversal, adrenal hypoplasia critical region, on chromosome X, gene-1; NR0B1) expressions [7], two well-known negative regulators of steroidogenesis. This is in turn is associated with germ cell loss and altered male fertility [7].

In addition to FXRα, bile acids are ligands for the G-protein-coupled membrane bile acid receptor 1 (GPBAR1; TGR5). While the role of TGR5 in the testis has been explored in relation to the regulation of germ cell homeostasis [8,9,10], no study has thus far analyzed its potential impact on endocrine Leydig cells. The present study, using in vitro approaches, deciphers novel roles for TGR5 to control Leydig cell homeostasis.

## 2. Results

### 2.1. TGR5 Was Expressed in Mouse Leydig Cells

To define whether TGR5 is expressed in Leydig cells, we carried out several approaches. First, using RT-PCR analysis, *Tgr5* mRNA was detected in the interstitial compartment of the mouse testis (Figure 1A). Then, we performed primary Leydig cell culture experiments. The data show that *Tgr5* mRNA was expressed in adult mouse Leydig cells (Figure 1A). These data were supported by the detection of *Tgr5* mRNA in the murine immortalized Leydig cell line mLTC1 (murine Leydig tumor cell line 1) (Figure 1A). Combined, these data demonstrated that *Tgr5* mRNA was expressed in mouse Leydig cells.

### 2.2. TGR5 mRNA Expression Was Is Regulated by the LH/CG Signaling Pathway in Mouse Leydig Cells

Leydig cell homeostasis is regulated by the hypothalamus–pituitary axis through LH/CG (Luteinizing hormone/choriogonadotropin) signaling [11]. The present data show that mLTC1 cells responded to hCG with the increase in testosterone level as soon as 3 h after treatment (Figure 1B). Regarding gene expression regulations, hCG treatment led, as expected [12], to an upregulation of *Star* mRNA and a downregulation of *Lhcgr* mRNA accumulations (Figure 1C). The present results also confirm that hCG treatment led to lower mRNA accumulations of *Dax-1* and *Shp* (Figure 1C), two nuclear receptors known to repress steroidogenesis. These two receptors have been demonstrated to be target genes of the LH/CG pathway [7,13] and of FXRα within the Leydig cells [6,7]. Moreover, consistently with previous reports [12], hCG treatment led to an increase in *Fxrα* mRNA accumulation (Figure 1C). The mRNA accumulation of *Tgr5* was modulated by hCG. Indeed, hCG treatment led to an increase in *Tgr5* mRNA accumulation in short-term experiments from 1 h up to 6 h after exposure. Then, at 9 h and up to 24 h after hCG treatment, *Tgr5* mRNA levels were lowered compared to vehicle (NaCl 0.9%)-treated cells (Figure 1C). Interestingly, the mRNA expression of *Tgr5* was affected by hCG in a faster manner (starting at 1 h) than the one of *Fxrα*, which was affected only after 6 h. As bile acids have been demonstrated to alter testosterone metabolism, the present data suggest that TGR5 could play a role in the Leydig cells, which led us to explore the roles of TGR5 within Leydig cells.

### 2.3. TGR5 Activation Regulated Testosterone Production in mLTC1 Cells

To define the role of TGR5 in mouse Leydig cells, we explored its impact on endocrine function by measuring the production of testosterone using the mLTC1 cell line. As TGR5 modulates the protein kinase (PKA) pathway [10,14,15], which is a fast second cellular messenger, we decided to analyze the effect of TGR5 activation in short-time experiments. Treatment of mLTC1 cells with the TGR5 agonist INT-777 for 3 h resulted in a significant decrease in intracellular testosterone levels compared to vehicle-treated cells, while this effect was not observed after 6 h (Figure 2A). A similar decrease in testosterone levels was observed between the culture media of vehicle-treated cells and those of INT-777-treated cells 3 h after treatment (Figure 2B). To circumvent the possibility that immortalized cell lines have a reduced ability to produce testosterone compared to normal Leydig cells, we also measured progesterone, an intermediate in testosterone synthesis. INT-777 treatment was also associated with lower levels of progesterone compared to vehicle-treated cells (Figure 2B). These data support the impact of TGR5 on steroid synthesis.

No impact on the mRNA accumulations of steroidogenic genes such as *Star, Cyp11a1, Hsd3b1*, or *Cyp17a1* was observed (Figure 2C). In addition, no impact of INT-777 was observed on the mRNA accumulations of *Mrp4* and *Mrp3* (Figure 2D), transporters known to modulate testosterone homeostasis in Leydig cells [16,17]. Note that the other transporter *Mrp2* was not detected in mLTC1 (data not shown). These results suggest that there was no impact of INT-777 on testosterone export.

TGR5 is known to act through the downstream PKA pathway [10,15,18]. To ensure the involvement of the PKA signaling pathway in response to INT-777, experiments using H89, a selective and potent PKA inhibitor, were performed. As expected, H89 led to a lower level of CREB-phosphorylation (Appendix A). The present data highlight the major role of PKA in the mechanism explaining how TGR5 regulates testicular steroidogenesis in Leydig cells. Indeed, the decrease in testosterone levels observed after INT-777 exposure was not detected in cells pretreated with H89 for 2 h, as revealed by intracellular and medium levels of testosterone (Figure 2E). Consistently with the negative impact of INT-777 on steroid synthesis in mLTC1 cells, INT-777 treatment also led to a decrease in CREB-phosphorylation compared to vehicle-treated cells (Figure 2F).

TGR5 shares the same downstream pathways as LH/CG, namely, PKA, which regulates the expression of steroidogenic genes. According to the present results, TGR5 and LH/CG modulate PKA-CREB signaling in an opposite way to mLTC1 cells. We wondered whether activation of TGR5 could modulate the sensitivity of the Leydig cells to the LH/CG pathway. In our hands, no impact on *Lhcgr* mRNA accumulation was noticed following TGR5 activation (Figure 3A). Consistently, no impact of the pre-treatment to INT-777 for 24 h was observed on hCG-induced testosterone levels in mLTC1 cells (Figure 3B). Combined, these data suggest that TGR5 must be a local regulator of testosterone synthesis that does not modulate the central regulation by LH/CG.

### 2.4. TGR5 Did Not Regulate Glucose Homeostasis in Mouse Leydig Cells

Glucose-6-phophatase (G6pase) and phosphoenol-pyruvate-kinase (Pepck), two enzymes involved in glucose homeostasis, have been shown to regulate the endocrine function of the Leydig cells [19]. In recent years, it has been reported that TGR5 controls glucose metabolism [20]. We wondered whether the impact of TGR5 on testosterone levels could be through an effect on metabolic pathways within the Leydig cells. We thus analyzed whether TGR5 activation could lead to abnormal glucose metabolism in murine Leydig cells. The present data show that activation of TGR5 by INT-777 did not modulate glucose levels in mLTC1 cells (Figure 4). These data suggest that TGR5 might not act through the modulation of glucose homeostasis to alter steroid production.

### 2.5. TGR5 Regulated Lipid Homeostasis in mLTC1 Mouse Leydig Cells

Some regulatory links have been reported between lipid metabolism and TGR5 [21,22]. Among lipids, cholesterol and triglycerides have major impacts on the endocrine capacities of Leydig cells [23,24]. Alteration of lipid metabolism could be visualized in steroidogenic cells using Nile-Red-O staining. We performed staining on mLTC1 treated with vehicle or INT-777 (Figure 5), and it was difficult to define clear data from such analysis as Nile-Red-O staining reveals neutral lipids, triglycerides, and cholesterol esters stored in lipid droplets. No difference was observed on the number of lipid droplets (Figure 5). However, a slight increase in the size of lipid droplets was detected in cells treated with INT-777 compared to vehicle-treated cells (Figure 5). However, non-common variation in the concentrations of either triglycerides or cholesterol esters could lead to misinterpretations.

We then decided to analyze TG levels. The present data show that the activation of TGR5 by the INT-777 had a significant negative impact on the intracellular triglyceride levels (Figure 6A). However, the deregulation of the metabolites was not associated with the alteration of the mRNA accumulations of genes involved in TG synthesis such as *Acc (acetyl-CoA carboxylase), Fas (fatty acid synthase),* or *Srebp1c (sterol-regulatory-element-binding protein 1)* or other genes such as *Dgat1 (diacylglycerol O-acyltransferase*) and *Dgat2,* for example (Figure 6B). 

As no alteration was observed at the mRNA levels, we decided to analyze important actors at the protein level. Indeed, it was demonstrated that the phosphorylation of ACC is key to control its activity (Figure 6C). The data show that the levels of ACC and P-ACC were decreased following INT-777 exposure (Figure 6C).

The effect of INT-777 on TG levels was abolished by pre-treatment with H89 (Figure 6D). These data highlight the complex association between the TGR5 signaling pathway and triglyceride homeostasis. This is consistent with previous data showing the impact of activation of TGR5 signaling in the regulation of hepatic triglyceride homeostasis, which contributes to protection against non-alcoholic fatty liver disease (NAFLD), although the molecular mechanisms have not been elucidated [25].

Even though there was a decrease in TG levels, a slight increase in lipid droplet size was observed in INT-777-treated cells (Figure 5), suggesting that there might be an accumulation of other lipid droplet components, such as cholesterol esters. We thus decided to analyze cholesterol levels, which is the initial substrate for steroidogenesis. The data show that the activation of TGR5 by the INT-777 led to an increase in total cellular cholesterol content (Figure 7A). This was not associated with the modulations of the mRNA accumulations of genes involved in their synthesis such as *HmgcoA-reductase or HmgcoA-synthase* (Figure 7B). The altered size of lipid droplets in INT-777-treated cells suggested that there may be an alteration in cholesterol storage. We analyzed the levels of cholesterol esters. The data show that this increase in cholesterol level corresponded to an increase in cholesterol ester levels (Figure 7A). These data suggest that there might be an increased storage or an alteration of the hydrolysis of cholesteryl esters stored in lipid droplets, which could explain both high cholesterol contain and lower testosterone production. For that, we analyzed by RT-qPCR the mRNA accumulation of the *hormone-sensitive lipase* (*Hsl*) that is responsible for neutral cholesteryl ester hydrolase activity. No modulation of the mRNA accumulation of *Hsl* was observed in response to INT-777 (Figure 7B). The effect of the INT-777 on cholesterol levels was abolished by pre-treatment with H89 (Figure 7C).

These data suggest that altered lipid homeostasis induced by TGR5 in Leydig cells could play a role in the lower capacity of these cells to produce testosterone, probably through the alteration of the availability of the substrate for steroidogenesis. However, the exact molecular mechanisms still need to be deciphered.

### 2.6. TGR5 Regulated Testosterone Production in Mouse Primary Leydig Cells

Since mLTC-1 cells are derived from tumor, and the fact that lipid homeostasis, and particularly cholesterol, is regulated differently between tumoral and normal cells, we decided to assess whether the impacts of INT-777 were reproducible on mouse primary Leydig cells.

Surprisingly no effect of INT-777 on TG was observed in mouse Leydig primary cell cultures *(*Figure 8A). In contrast, the data show that the impact of INT-777 on cholesterol ester levels was confirmed using primary Leydig cell cultures (Figure 8B). In addition, the experiments performed on mouse Leydig primary cells validated the negative impact of INT-777 treatment on steroid production, as revealed by the measurements of progesterone and testosterone levels in cells and medium (Figure 8C).

## 3. Discussion

The role of the bile acid receptor TGR5 in the testis has been previously demonstrated in germ cell lineage [8,9,10]. However, its roles on the testicular endocrine cells have not been studied so far. The present work demonstrates the expression of *Tgr5* mRNA in mouse Leydig cells. This led us to study what could be the cellular function regulated by TGR5 in Leydig cells.

We provide evidence for the negative impact of TGR5 activation using INT-777 on the synthesis of testosterone. This effect did not seem to rely on the regulation at the mRNA level of the expression of genes involved in steroidogenesis. However, the steroidogenic pathway has been shown to be regulated by post-translational modifications of key proteins, as demonstrated for STAR [26]. Indeed, it appeared that the phosphorylation of STAR intervenes to support its maximum activity for the transport of cholesterol into the mitochondria and thus to initiate the synthesis of steroids [26]. Thus, the impact of the TGR5 signaling pathway on post-transcriptional and/or post-translational effect must be analyzed to decipher the mechanisms involved before assuming that the effect is through mechanisms independent of gene expression regulation.

Even though it has not been shown before, this involvement of TGR5 in the regulation of steroidogenesis is conceivable. Indeed, it is well known that TGR5 mainly acts through the activation of the adenylate cyclase, leading to the activation of the PKA pathway and by the end to the phosphorylation of CREB [27]. This PKA–CREB pathway has been demonstrated for decades to control the production of testosterone in response to the LH/CG signal [28,29]. Surprisingly, if in the context of LH/CG stimulation, the mobilization of the PKA pathways leads to higher testosterone production, it was unexpected that TGR5 activation could led to lower testosterone levels through PKA, as suggested by the present experiments using H89. The present data suggest that in the mouse Leydig cells, TGR5 activation leads to a decrease in CREB phosphorylation. This is quite unexpected as in other cell types, TGR5 activation was mainly associated with an increase in CREB phosphorylation [10,22]. Further studies are needed to better decipher the molecular mechanisms explaining why in Leydig cells TGR5 acts in an opposite way on PKA-CREB pathway.

In addition, even if the LH/CG and TGR5 pathways share common downstream pathways, the present results suggest that there is no interference by co-treatment with INT-777 and LH/CG. It thus appears that TGR5 pathway only regulates the basal production of testosterone and not the LH/CG induced production. However, it remains to be deciphered as to what could be the mechanisms by which the TGR5-PKA pathway leads to a lower testosterone level. 

In recent years, TGR5 has been demonstrated to be involved in the control of several physiological functions [22]. TGR5 controls glucose and lipid metabolisms, and these two metabolisms are known to be involved in the homeostasis of steroid synthesis [19,23]. However, the present data suggest that TGR5 activation has no impacts on glucose levels in Leydig cells. In contrast, within Leydig cells, TGR5 activation led to an alteration of lipid droplets (cholesterol esters and triglycerides), which are a reservoir of substrate for the synthesis of steroids [17]. Using the mLTC1 cell line and mouse primary Leydig cells, it appears that activation of TGR5 by INT-777 leads to an increase in cellular cholesterol content, mainly cholesterol ester levels. This alteration of the lipid metabolism induced by INT-777 could participate in the drop in testosterone levels in the Leydig cells.

The present data highlight the impact of TGR5 signaling pathway on testosterone production. These data must be of importance and require further studies as endocrine homeostasis is involved in the control of many physiological processes, and its alteration is associated with the development of diseases. In addition, the finding that TGR5 activation modulates the steroid production has to be put in line with the actual questioning of the impacts of environmental endocrine disruptors in the programming of health and diseases [30,31,32].

In the present study, the concentration of INT-777 used (25 μM) is consistent with other published works using this compound [15]. The issue of bile acid concentrations within the testicles has been addressed in the context of hepatic pathologies. It has thus been clearly defined that the testicular concentrations of bile acids observed in this context of hepatic pathologies are compatible with an activation of TGR5. Thus, it will be necessary to better define under which conditions (physiological or pathological) the levels of testicular bile acids will be sufficient to activate the TGR5 in the Leydig cells leading to lower testosterone levels.

Another remaining question is to decipher the endogenous testicular ligand of TGR5. Indeed, if bile acids have been demonstrated to reach the testis from blood [9] and that testis could even produce some bile acids [33], we could not exclude that other endogenous molecules could act as testicular TGR5 agonists. Indeed, some steroids have been demonstrated to activate TGR5, such as pregnandione [34]. This clearly supports the idea of a crosstalk between TGR5 signaling pathway and testicular steroidogenesis.

In the present work, we point out that the G-protein-coupled bile acid receptor TGR5 is expressed in the Leydig cells, where its expression is controlled by the main regulator of the Leydig cell homeostasis, the LH/CG signaling pathway. The regulation of *Tgr5* mRNA accumulation by hCG is dynamic as it is first induced in a short-term experiment and then repressed. This is quite interesting as a similar expression pattern was demonstrated for the mRNA expression of the nuclear bile acid nuclear receptor *Fxrα* [12] for the regulation of the testicular endocrine function [5]. Indeed, in recent years, bile acid homeostasis has been associated with the modulation of the testosterone production by Leydig cells [4]. This is consistent with the fact that in cholestasis disease, testosterone levels are lower than in normal physiological conditions. However, until now, only the nuclear bile acid receptor FXRα was demonstrated to be involved [5,12]. This is consistent with the fact that FXRα pathways repress the expression of steroidogenic genes such as *Star* [6,7]. Moreover, it has been demonstrated that FXRα pathways also regulate the sensitivity of Leydig cells to the LH/CG, connecting the central regulation of steroidogenesis to local tight control [6,7]. The present data highlight the role of TGR5 in the regulation of the testicular endocrine function. Interestingly, the kinetic of the regulation could be of importance as TGR5 seems to act rapidly on steroidogenesis (3 h, shown herein), whereas for FXRα, the effect was seen on Leydig cells in a longer time frame (12 to 24 h) [6]. This could reflect the differences in the mechanisms of action between a G-protein-coupled receptor versus nuclear receptor. Overall, these data sustain that bile acids either through TGR5 and FXRα exert a negative effect on testicular androgen synthesis. It is also interesting to note that the LH/CG pathway leads first to an upregulation of the expression of both *Tgr5* and *Fxrα*. It is conceivable that the upregulation of their expression levels in response to LH/CG is a way to initiate a kind of negative feedback. However, it is interesting to note that LH/CG modulates the expression of *Tgr5* more rapidly than that of *Fxrα*. We could thus hypothesize that TGR5 could be “in the first line” to ensure the feedback of testosterone synthesis, and then a second wave arrives through FXRα signaling, which takes longer via the regulation of *Shp* and *Dax-1* expressions.

In addition to the established role of FXRα in the Leydig cells, the identification of TGR5 signaling as a regulator of testicular steroidogenesis, with a similar negative effect on steroid synthesis, opens new perspectives to better understand the complex impacts of bile acids on Leydig cell homeostasis and on the testicular endocrine function. 

Our data reinforce the links between the bile acid signaling pathways through TGR5 and/or FXRα and the control of the endocrine function of the Leydig cells. Combined with previous data, the present results define TGR5 and FXRα as critical actors that need to be more deeply explored to increase our knowledge of pathologies associated with altered testicular endocrine function such as fertility disorders.

## 4. Materials and Methods

### 4.1. Animals

C57Bl/6J mice were purchased from Charles River Laboratories (L’Arbresle, France). Mice were acclimated at least 2 weeks before experiments. Mice were housed in temperature-controlled rooms with 12 h light/dark cycles. Mice had ad libitum access to food and water. The refinement is based on the housing and monitoring of the animals as well as the development of protocols that consider animal welfare. This has been achieved by enriching the cages (cardboard tunnel and mouse houses). This study was conducted in accordance with current regulations and standards approved by Institut National de la Santé et de la Recherche Médicale Animal Care Committee and by the animal care committee.

### 4.2. Interstitial Cell Enrichment

For the generation of data on enriched interstitial cells compared to tubular preparations or whole testis samples, the methodology used relies on mechanical and enzymatic processes for dissociating the fractions as previously described [12]. The interstitial enriched cells and tubular compartments were generated as described in a previous study. Briefly, testes from 90-day-old male mice were decapsulated and incubated for 20 min at 33 °C in Dulbecco’s modified Eagle’s medium (DMEM)/Ham’s F12 (1:1), transferrin (5 μg/mL), insulin (4 μg/mL), and vitamin E (0.2 μg/mL) medium containing collagenase (0.8 mg/mL) (Life Technologies, Invitrogen, Cergy-Pontoise, France). Extracts were collected by centrifugation for 10 min at 200× *g*, and the pellet was resuspended in fresh medium.

### 4.3. Cell Line Approaches

mLTC1 cells were used as previously described [35]. Cells were plated in 12-well plates. Twenty-four hours after plating, cells were put in serum-free medium overnight. Then, cells were treated with vehicles (DMSO 1/1000 (for H89) or NaCl0.9% 1/1000 (for hCG)), INT777 (25 µM; Sigma-Aldrich, St. Louis, MO, USA), or hCG (25 nM, Sigma-Aldrich, St. Louis, MO, USA). Following this, cells were harvested at different time points, and messenger RNA (mRNA) or protein extractions were performed.

### 4.4. Real-Time RT-qPCR

RNA from cell samples were isolated using RNAzol. cDNA was synthesized from total RNA with the MMLV and random hexamer primers (Promega, Charbonnières-les-Bains, France). The real-time PCR measurement of individual cDNA was performed using SYBR green dye (Master mix Plus for SYBR Assay, *Takara Bio*) to measure duplex DNA formation with the Eppendorf–Realplex system. For each experiment, standard curves were generated with pools of cDNA from cells with different genotypes and/or treatments. The results were analyzed using the ΔΔct method. Primer sequences are reported in Table 1.

### 4.5. Nile-Red-O Staining

Cells were plated in 6-well plates. Twenty-four hours after plating, cells were put in serum-free medium overnight. Then, cells were treated for with vehicle (DMSO 1/1000) or INT777 (25 µM; Sigma-Aldrich, St. Louis, MO, USA). Three hours after treatment, cells were fixed with 4% PFA fixation (10 min room temperature agitation). Cells were washed 3 times for 5 min with PBS-Tween 0.01% at room temperature. Then, saturation was performed for 1 h at room temperature with PBS-Tween 0.01% + BSA. Following this, cells were incubated in PBS-Tween 0.01% with Nile-Red-O (1 ng/mL) for 30 min. Four washes were performed with PBS. Cells were then counterstained with Hoechst medium (1 mg/mL) and then mounted on PBS/glycerol (50/50).

### 4.6. Glucose, Cholesterol, Cholesterol Ester, and Triglyceride Measurements

Cells were plated in 6-well plates. Twenty-four hours after plating, cells were put in serum-free medium overnight. Then, cells were treated for with vehicle (DMSO 1/1000 (for H89) or NaCl0.9% 1/1000 (for hCG)), INT777 (25 µM; Sigma-Aldrich, St. Louis, MO, USA), or hCG (25 nM, Sigma-Aldrich, St. Louis, MO, USA). Following this, cells were harvested in PBS1X. After this, measurements conducted using colorimetric assays were performed on cells or medium following the recommendations of the manufacturer (Glucose RTU, 61269, Biomerieux SA, France). Triglyceride measurements were performed using a colorimetric assay kit (Triglycerides FS*, DiaSys Diagnostic Systems GmbH, Holzheim, Germany).

### 4.7. Testosterone Measurements

Cells were plated in 6-well plates. Twenty-four hours after plating, cells were placed in serum-free medium overnight. Then, cells were treated with vehicles (DMSO 1/1000 (for H89) or NaCl0.9% 1/1000 (for hCG)), INT777 (25 µM; Sigma-Aldrich, St. Louis, MO, USA), or hCG (25 nM, Sigma-Aldrich, St. Louis, MO, USA). Following this, cells were harvested at different time points. Testosterone was measured in the cell extract of MLTC1 cells as previously described using a commercial kit (AR-K032-H5, ARBOR ASSAYS, INC), which was used for the assays.

### 4.8. Statistical Analyses

All numerical data are represented as mean ± SEM. Significant difference was set at *p* < 0.05. Differences between groups were determined by *t*-test.

## Figures and Tables

**Figure 1 ijms-23-15398-f001:**
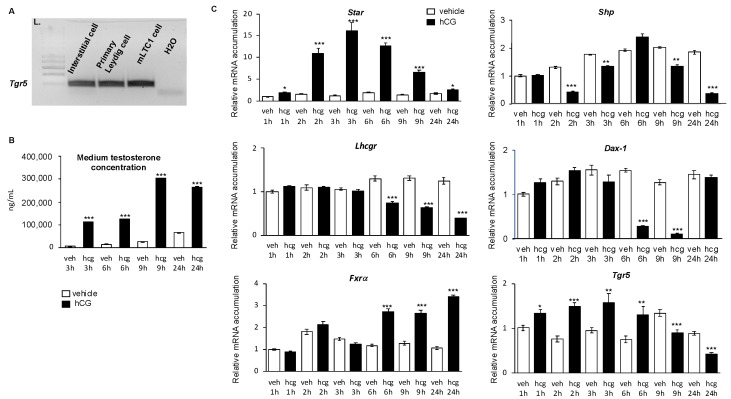
*Tgr5* was expressed in mouse testis. (**A**) Detection of *Tgr5 mRNA* in mouse testicular interstitial compartment, in mouse primary Leydig cells, and in the mLTC1 cell line determined by RT-PCR analysis. (**B**) Testosterone concentrations determined in the culture medium of mLTC1 cells treated with vehicle (NaCl 0.9%) or hCG (25 nM) for 3, 6, 9, or 24 h. (**C**) Relative *Star, Shp, Lhcgr, Dax-1, Fxrα,* and *Tgr5* mRNA accumulations normalized to *β-actin* in mLTC1 cells treated with vehicle (NaCl 0.9%) or hCG (25 nM) for 1, 2, 3, 6, 9, or 24 h. In all panels, n = 9 to 12. Data are expressed as the means ± SEM. Statistical analysis: *, *p* < 0.05; **, *p* < 0.01; ***, *p* < 0.001 versus vehicle-treated cells for the same timing.

**Figure 2 ijms-23-15398-f002:**
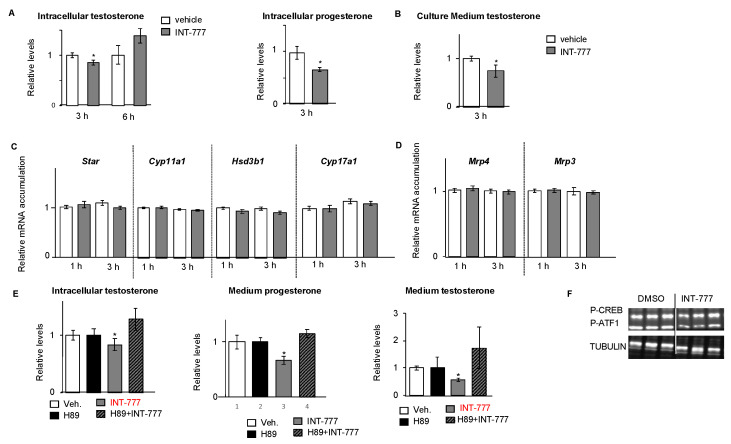
TGR5 regulated testosterone metabolism in a mouse Leydig cell line. (**A**) Relative intracellular levels of testosterone and progesterone normalized to protein concentrations in mLTC1 cells treated for 3 h or 6 h with vehicle or INT-777. (**B**) Relative levels of testosterone in culture medium, normalized to protein concentration, in mLTC1 cells treated for 3 h with vehicle or 25 mM INT-777. (**C**) Relative *Star, Cyp11a1, Hsd3b1*, and *Cyp17a1* mRNA accumulations normalized to *β-actin* in mLTC1 cells treated for 1 or 3 h with vehicle or 25 mM INT-777. (**D**) Relative *Mrp4* and *Mrp3* mRNA accumulations normalized to *β-actin* in mLTC1 cells treated for 1 or 3 h with vehicle or INT-777. (**E**) Relative intracellular levels of testosterone, normalized to protein concentration, in mLTC1 cells pre-treated with H89 for 2 h and then for 3 h with vehicle or INT-777. (**F**) Representative Western blot of P-CREB and TUBULIN in mLTC1 cells treated with vehicle or INT-777. For quantification, vehicle treated cells were arbitrarily set at 1. In all panels, n = 9 to 12. Data are expressed as the means ± SEM. Statistical analysis: *, *p* < 0.05 compared to vehicle treated cells.

**Figure 3 ijms-23-15398-f003:**
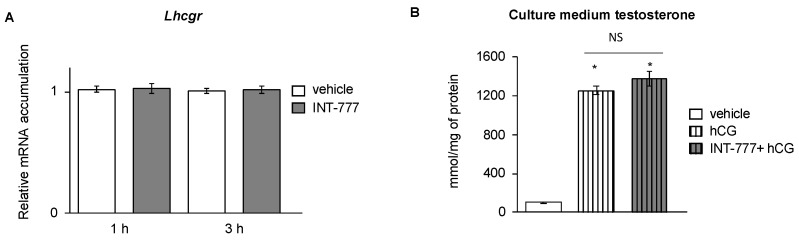
TGR5 did not regulate sensitivity of mLTC1 cells to hCG. (**A**) Relative *Lhcgr* mRNA accumulation normalized to β-actin in mLTC1 cells treated with vehicle or INT-777 for 1 h or 3 h (**B**) Relative intracellular levels of testosterone, normalized to protein concentration, in mLTC1 cells pre-treated with vehicle (DMSO) or INT-777 for 24 h and then for 2 h with vehicle (NaCl 0.9%) or hCG. n = 9 to 12. Data are expressed as the means ± SEM. Statistical analysis: * *p* < 0.05 compared to vehicle treated cells. NS: non-significant.

**Figure 4 ijms-23-15398-f004:**
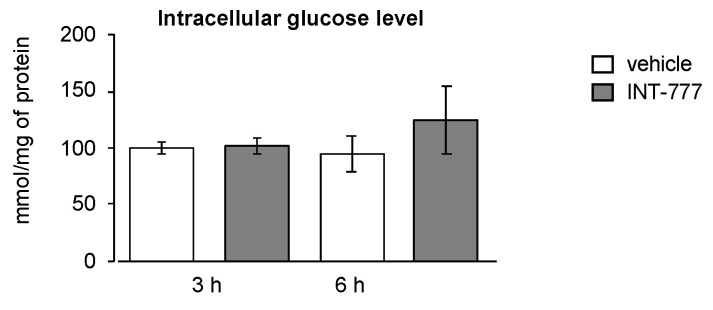
TGR5 did not regulate glucose metabolism in the mouse Leydig cell line. Relative intracellular levels of glucose in mLTC1 cells normalized to protein concentration 3 h after treatment with vehicle (DMSO) or INT-777. n = 9 to 12. Data are expressed as the means ± SEM.

**Figure 5 ijms-23-15398-f005:**
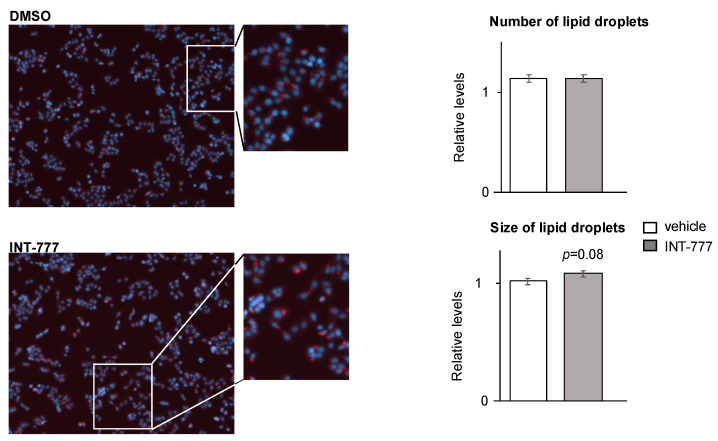
TGR5 modulated lipid metabolism in a mouse Leydig cell line. Representative images mLTC1 cells 3 h after treatment with vehicle (DMSO) or INT-777 and stained with Nile-Red-O. Quantifications of the number and size of lipid droplets in mLTC1 cells 3 h after treatment with vehicle (DMSO) or INT-777 and stained with Nile-Red-O. In all panels, n = 9 to 12. Data are expressed as the means ± SEM.

**Figure 6 ijms-23-15398-f006:**
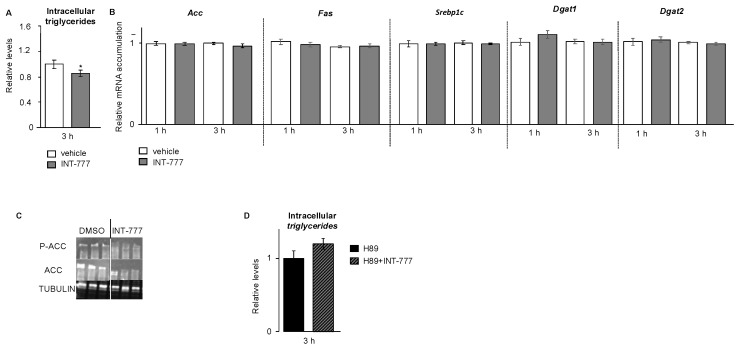
TGR5 modulated TG metabolism in a mouse Leydig cell line. (**A**) Relative intracellular levels of cholesterol in mLTC1 cells normalized to protein concentration 3 h after treatment with vehicle or INT-777. (**B**) Relative mRNA accumulations of *Acc*, Fas, *Srepb1c*, *Dgat1,* and *Dgat2* normalized to β-actin in a mouse Leydig mLTC1 cell line treated with vehicle or hCG for 3 h. (**C**) Representative Western blots of P-ACC, ACC, and TUBULIN in mLTC1 cells treated with vehicle (DMSO) or INT-777 for 3 h. (**D**) Relative intracellular levels of cholesterol in mLTC1 cells normalized to protein concentration in cells pre-treated 2 h with H89 and then 3 h with vehicle or INT-777. In all panels, n = 9 to 12. Data are expressed as the means ± SEM. Statistical analysis: * *p* < 0.05.

**Figure 7 ijms-23-15398-f007:**
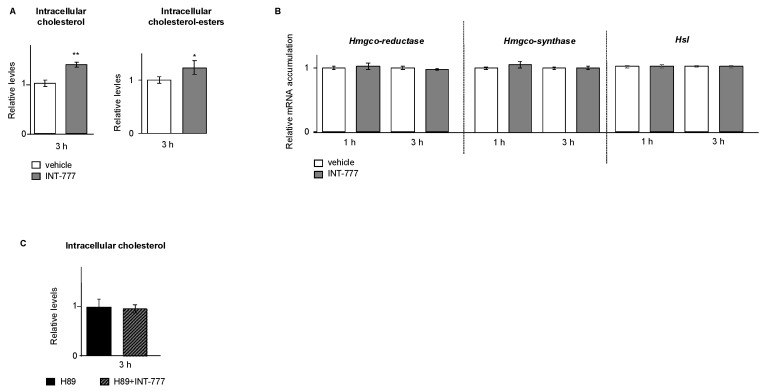
TGR5 regulated triglyceride homeostasis in a mouse Leydig cell line. (**A**) Relative intracellular levels of cholesterol and cholesterol esters in mLTC1 cells normalized to protein concentration 3 h after treatment with vehicle or INT-777. (**B**) Relative mRNA accumulations of *HmgcoA-reductase*, *HmgcoA-synthase,* and *Hsl* normalized to *β-actin* in a mouse Leydig mLTC1 cell line treated with vehicle or hCG for 3 h. (**C**) Relative intracellular levels of cholesterol in mLTC1 cells normalized to protein concentration in cells pre-treated 2 h with H89 and then 3 h with vehicle or INT-777. In all panels, n = 9 to 12. Data are expressed as the means ± SEM. Statistical analysis: * *p* < 0.05, ** *p* < 0.01.

**Figure 8 ijms-23-15398-f008:**
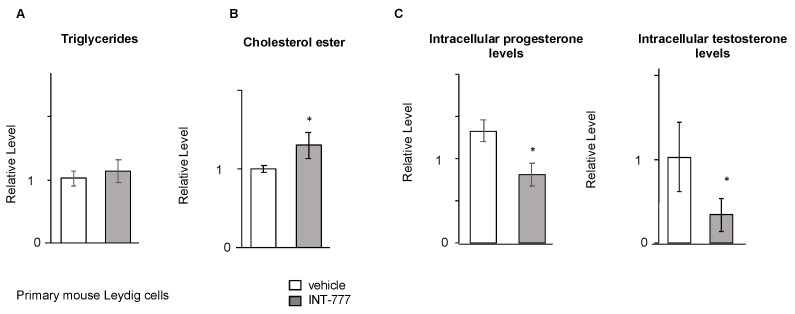
TGR5 signaling was effective in mouse primary Leydig cells. (**A**) Relative intracellular levels of triglycerides in mouse primary Leydig cells normalized to protein concentration 3 h after treatment with vehicle or INT-777. (**B**) Relative intracellular levels of cholesterol esters in mouse primary Leydig cells normalized to protein concentration 3 h after treatment with vehicle or INT-777. (**C**) Relative intracellular levels of progesterone and testosterone in mouse primary Leydig cells normalized to protein concentration 3 h after treatment with vehicle or INT-777. In all panels, n = 3 to 6. Data are expressed as the means ± SEM. Statistical analysis: * *p* < 0.05.

**Table 1 ijms-23-15398-t001:** Sequences of primers used in this study.

Gene Name	Forward	Reverse
* **Actin** *	TCATCACTATTGGCAACGAGC	AGTTTCATGGATGCCACAGG
* **Lhcgr** *	AGCTAATGCCTTTGACAACC	GATGGACTCATTATTCATCC
* **Star** *	TGTCAAGGAGATCAAGGTCCT	CGATAGGACCTGGTTGATGAT
* **Cyp11a1** *	CTGCCTCCAGACTTCTTTCG	TTCTTGAAGGGCAGCTTGTT
* **Hsd3b1** *	ATGGTCTGCCTGGGAATGAC	ACTGCAGGAGGTCAGAGCT
* **Cyp17a1** *	CCTGATACGAAGCACTTCTCG	CCAGGACCCCAAGTGTGTTCT
* **Shp** *	CGATCCTCTTCAACCCAGATG	AGGGCTCCAAGACTTCACACA
* **Dax-1** *	GTCCAGGCCATCAAGAGTTTC	CAGCTTTGCACAGAGCATCTC
* **Acc** *	GCCTTTCACATGAGATCCAGC	CTGCAATACCATTGTTGGCGA
* **Fas** *	AAGCGGTCTGGAAAGCTGAA	AGGCTGGGTTGATACCTCCA
* **Srebp1c** *	GAACAGACACTGGCCGAGAT	GAGGCCAGAGAAGCAGAAGAG
* **HmgcoA-reductase** *	ACAGAAACTCCACGTGACGA	TTCAGCAGTGCTTTCTCCGT
* **HmgcoA-Synthase** *	GCAAAAAGATCCGTGCCCAG	GTCATTCAGGAACATCCGAGC
* **Mrp3** *	AGTCTTCGGGAGTGCTCATCA	AGGATTTGTGTCAAGATTCTCCG
* **Mrp4** *	TTAGATGGGCCTCTGGTTCT	GCCCACAATTCCAACCTTT
* **Dgat1** *	TGGCTGCATTTCAGATTGAG	GCTGGGAAGCAGATGATTGT
* **Dgat2** *	CTTCCTGGTGCTAGGAGTGGC	GCTGGATGGGAAAGTAGTCTCGG
* **Hsl** *	CATATCCGCTCTCCAGTTGACC	CCTATCTTCTCCATCGACTACTCC

## Data Availability

Data are available upon request to corresponding author.

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
