# Peer review of "Identification of the Role of TGR5 in the Regulation of Leydig Cell Homeostasis"

_ijms, 2022, doi:10.3390/ijms232315398_

Round 1

Reviewer 1 Report

This manuscript addresses the roles of TGR5 on testosterone production and lipid metabolism in a Leydig cell line. The activation of TGR5 by INT-777 decreased the production of testosterone. Since the activation of TGR5 decreased TG and increased cholesterol concentrations in the cells, they concluded that altered lipid homeostasis (the availability of the substrate for steroidogenesis) could be a mechanism explaining lower testosterone levels induced by TGR5. 

1.  As shown in Fig. 2, the inhibitory effect of INT-777 (25 µM) on testosterone production is slight. In addition, the concentration of added INT-777 is much higher than that of TGR5 ligand secondary bile acids in peripheral blood. They need to discuss the possibility that serum or intratesticular bile acids affect lipid metabolism in Leydig cells via TGR5 in terms of ligand concentration.

2.  In Fig.5, the authors speculated that the reduced availability of the substrate (cholesterol) for steroidogenesis was a reason for the reduced biosynthesis of testosterone via TGR5. However, their data are too preliminary to discuss the mechanism of testosterone reduction by TGR5 activation. Is the increased intracellular cholesterol free cholesterol or esterified cholesterol? How does activation of TGR5 affect cholesterol ester hydrolysis?

3.  Is it possible that INT-777 could decrease testosterone concentration by increased testosterone catabolism (due to the activation of PXR or CAR)?

4.  Since FXRbeta (NR1H5) is a pseudogene, FXRalpha (NR1H4) is often referred to simply as FXR. Is there any reason to say FXRalpha instead of FXR?

5.  In lines 70, 71, and 88, does NaCl mean saline solution or PBS? In line 316, however, the authors state vehicle (DMSO, 1/1,000).

6.  In lines 98-101, they say INT-777 decreased (-20%) testosterone level after 3 hours (Fig. 2A). However, Fig. 2A does not appear to show a 20% reduction, nor is it statistically significant.

7.  In Fig.2E, what is the difference between the two "vehicles"?

8.  In lines 165-166, Figure 5B, and Table1, gene names (ID) of HMG-CoA reductase and HMG-CoA synthase are Hmgcr and Hmgcs, respectively.

9.  In lines 172-173, please be consistent with the order in which full names and abbreviations are listed.

10.  In lines 331-334, please mention how to measure cholesterol concentrations.

11.  There are several typos. 

In line 60, “intertitial” should be “interstitial”.

In line 78, “1 hour” may be “3 hours”.

In line 83, “Consitently” should be “Consistently”.

In line 102, “Figure 2B” may be “Figure 2C”.

In line 105, “Figure 2C” may be “Figure 2B”.

Author Response

Comments and Suggestions for Authors

This manuscript addresses the roles of TGR5 on testosterone production and lipid metabolism in a Leydig cell line. The activation of TGR5 by INT-777 decreased the production of testosterone. Since the activation of TGR5 decreased TG and increased cholesterol concentrations in the cells, they concluded that altered lipid homeostasis (the availability of the substrate for steroidogenesis) could be a mechanism explaining lower testosterone levels induced by TGR5. 

 We would like to thank this reviewer for his/her comments that improve the overall quality of our manuscript.

  1. As shown in Fig. 2, the inhibitory effect of INT-777 (25 µM) on testosterone production is slight. In addition, the concentration of added INT-777 is much higher than that of TGR5 ligand secondary bile acids in peripheral blood. They need to discuss the possibility that serum or intratesticular bile acids affect lipid metabolism in Leydig cells via TGR5 in terms of ligand concentration.

We understand the comment of the reviewer. In the past we have been able to demonstrate that the bile acid signaling pathways are able to modulate the testicular physiology and pathophysiology. This reviewer is right that the dose used here is quite high. We have to say that we use classical approach using INT-777 as other authors known as reference in the field use INT-777 to concentration of 30mM (Perino et al. PMID: 34031591); what is consistent with our experiments. Please note that TGR5 is a GPCR that is activated by both conjugated and unconjugated primary and secondary bile acids. Indeed, in contrast to other BA receptors, TGR5 is activated by all known bile acids, regardless of their substitution pattern and state of conjugation (un-, taurine- or glycine-conjugated), although with varying levels of potency ranging from 0.29 to 36.7 μM (Sato J Med Chem. 2008;51:1831–41).  However, the question remains of consistence between doses and bile acid concentrations found in blood and testis.

Our previous data demonstrate that BA are found in mouse testis and can modulate their signaling pathways (Baptissart 2014). These testicular concentrations reflect the average levels in the whole testis (interstitial and tubular compartments). Note that in testis the bile acids coming from blood flow arrived by blood capillaries located in the interstitial space, where are located Leydig cells too. This suggests that Leydig cells could be exposed to high concentrations of BA corresponding.

However, to overcome the issue of the bile acid concentration within the testis in normal physiological conditions, we have added in the revised version of the manuscript a comment in the discussion part; where we highlight that the present results could found some relevance in the context of pathologies where the bile acid concentrations are elevated such as liver disorders (cholestasis, hepatitis, cirrhosis…. or even obesity).

  1. In Fig.5, the authors speculated that the reduced availability of the substrate (cholesterol) for steroidogenesis was a reason for the reduced biosynthesis of testosterone via TGR5. However, their data are too preliminary to discuss the mechanism of testosterone reduction by TGR5 activation. Is the increased intracellular cholesterol free cholesterol or esterified cholesterol? How does activation of TGR5 affect cholesterol ester hydrolysis?

We have added data on the quantity of cholesterol ester. These data highlight that the increase of cholesterol content observed following INT-777 treatment is due to elevated storage of cholesterol ester. Thus, the activation of TGR5 seems to affect storage of cholesterol as cholesterol esters rather than leading to hydrolysis.

The present data did not lead us to identify the involved mechanisms at the molecular level. But these data open perspectives to further study the role of TGR5 in the disruption of the endocrine function. What must be highly relevant in the context of the environmental exposure to endocrine disruptors.

  1. Is it possible that INT-777 could decrease testosterone concentration by increased testosterone catabolism (due to the activation of PXR or CAR)?

So far, INT-777 has been defined as a TGR5 specific agonist.

As confidential comments, we would like to add that we have unpublished data on PXR and CAR activation that are part of another story (so that cannot be published here), the profiles of the responses to PXR or CAR agonists are not at all consistent with what is observed using INT-777. As these results are parts of another paper I cannot said more here, if these point-by point is published, but if needed, we will be happy to discuss them with the academic editor.

  1. Since FXRbeta (NR1H5) is a pseudogene, FXRalpha (NR1H4) is often referred to simply as FXR. Is there any reason to say FXRalpha instead of FXR?

We put FXRa in the text because this is the official nomenclature of the nuclear bile acid receptor. We wanted to avoid potential misunderstanding between FXRbeta and FXRalpha. We thus have decided to let it that way throughout the text.

  1. In lines 70, 71, and 88, does NaCl mean saline solution or PBS? In line 316, however, the authors state vehicle (DMSO, 1/1,000).

Yes indeed, Nacl means saline solution (NaCl 0.9%); this has been specified in the text now.

In the experiments, there were two different vehicles as they served to dissolve different molecules. For each case, vehicle was always administrated at 1/1000.

One is NaCl 0.9% to be compared with hCG;

and one is DMSO to be compared with INT-777.

  1. In lines 98-101, they say INT-777 decreased (-20%) testosterone level after 3 hours (Fig. 2A). However, Fig. 2A does not appear to show a 20% reduction, nor is it statistically significant.

This reviewer is right. The mention (-20%) has been removed in the revised version.

  1. In Fig.2E, what is the difference between the two "vehicles"?

Sorry in this original Fig2E, there was an error the grey bar was in fact INT-777. It has been corrected in the revised version.

  1. In lines 165-166, Figure 5B, and Table1, gene names (ID) of HMG-CoA reductase and HMG-CoA synthase are Hmgcr and Hmgcs, respectively.

The gene names have been changes to be consistent between Figure 5B and Table 1.

  1. In lines 172-173, please be consistent with the order in which full names and abbreviations are listed.

 We check the text and have now corrected the text to be consistent with the order in which full names and abbreviations are listed.

  1. In lines 331-334, please mention how to measure cholesterol concentrations.

We added a section in the materials and methods to report how was performed the measurement of cholesterol concentrations.

  1. There are several typos. 

In line 60, “intertitial” should be “interstitial”. Change has been done.

In line 78, “1 hour” may be “3 hours”. Change has been done.

In line 83, “Consitently” should be “Consistently”. Change has been done.

In line 102, “Figure 2B” may be “Figure 2C”. Change has been done according to the revised version considering the added data.

In line 105, “Figure 2C” may be “Figure 2B”. Change has been done according to the revised version considering the added data.

Reviewer 2 Report

This study is a continuation of the work from this research group aiming to characterize the role of bile acids, FXRa, and now TGR5 in the testis. The results suggest that TGR5 plays a role in Leydig cell steroidogenesis via the regulation of lipid homeostasis.

Major comments:

1.      In the results section, the authors report that INT-777 significantly increased cholesterol levels in a mouse Leydig cell line. Since MLTC-1 cells are derived from a tumor and cholesterol synthesis is regulated differently between tumoral and normal cells, it will be interesting to assess whether INT-777 affects cholesterol levels in primary Leydig cells. Can the authors relate the changes in cholesterol levels (increase in response to INT-777) to changes in testosterone formation? Moreover, which cholesterol (cholesterol ester or free cholesterol) increases in response to INT-777?

2.      The authors also report that INT-777 significantly reduced TG levels, although the levels of the genes mediating this event did not change. How do the authors explain why TG levels changed? Are there other sources or changes at the protein levels of the enzymes mediating TG synthesis?

3.      In the discussion section, the authors mentioned that “TGR5 mainly acts through the activation of the adenylate cyclase, leading to the activation of the PKA pathway and by the end to the phosphorylation of CREB. This PKA-CREB pathway has been demonstrated for decades to control testosterone production in response to the LH/CG signal”. It will be critical to demonstrate this by examining whether TGR5 and the PKA agonist H89 affect CREB phosphorylation under these conditions. There are well-established methods and materials (antibodies) for such studies.

Minor comments:

4.      Please follow the guideline and nomenclature format for IJMS. For instance, gene names should be italicized in Figs 1, 2, 6, and throughout the main text. In Fig. 1A legend, PCR should be RT-PCR, and the same in the main text (Line 60).

5.      Other than the gene names in qPCR, the legend should not be italicized all through Fig. 2-6.

6.      In Line 78, should it be “3hr” instead of “1hr”?

7.      In Fig. 3B, to be consistent with the description in Line 122, it is suggested to add the “NS” between hCG and INT-77+hCG treatment. 

8.      The description of Figures 2B and 2C should be switched in lines 102 and 105.

9.      What is the difference between the “vehicle” in Fig. 2E?

10.   In line 134, “hr” should be added after the numbers 2 or 3.

11.   In Fig. 4, there is only one panel. Thus, “in all panels” in line 157 should be deleted.

12.   In Line 169, “slight” should be “significant”

13.   In Fig. 6C, “3hr” is suggested to be labeled.

14.   In line 203, “use” should be changed to “us”.

15.   In line 295, “mice” should be “Mice” starting a new sentence.

16.   Line 307, “2” at the end of the sentence should be deleted.

17.   In Table 1, the Tgr5 primers used should be included.

18.   To be consistent, the title should also include “TGR5” rather than “GPBAR1”.

Author Response

 Comments and Suggestions for Authors

This study is a continuation of the work from this research group aiming to characterize the role of bile acids, FXRa, and now TGR5 in the testis. The results suggest that TGR5 plays a role in Leydig cell steroidogenesis via the regulation of lipid homeostasis.

Major comments:

  1. In the results section, the authors report that INT-777 significantly increased cholesterol levels in a mouse Leydig cell line. Since MLTC-1 cells are derived from a tumor and cholesterol synthesis is regulated differently between tumoral and normal cells, it will be interesting to assess whether INT-777 affects cholesterol levels in primary Leydig cells. Can the authors relate the changes in cholesterol levels (increase in response to INT-777) to changes in testosterone formation? Moreover, which cholesterol (cholesterol ester or free cholesterol) increases in response to INT-777?

We have added data on the quantity of cholesterol ester in MTLC-1. These data highlight that the increase of cholesterol content observed following INT-777 treatment is due to elevated storage of cholesterol ester.

The reviewer is right, the homeostasis could be different between tumor and normal cells.

We performed Leydig primary culture cell experiments. Data confirm that even in normal Leydig cells, INT-777 leads to decrease testosterone levels as well as to an increase of cholesterol ester concentrations. These data sustained the fact that TGR5 is a regulator of steroid synthesis pathway in Leydig cells.

  1. The authors also report that INT-777 significantly reduced TG levels, although the levels of the genes mediating this event did not change. How do the authors explain why TG levels changed? Are there other sources or changes at the protein levels of the enzymes mediating TG synthesis?

This decrease in TG was not observed in the Leydig primary culture cell experiments. This support the idea of this reviewer suggesting that lipid homeostasis is regulated differently between tumoral and normal cells. This suggests that the alteration of TG in MLTC-1 is not of major impact considering the effect of TGR5 on testosterone levels as in primary cells there was the decrease of testosterone without impact on TG.

However, we analyzed the MLTC-1 cells at the protein level. The accumulation of the ACC and phospho-ACC are main actors of TG homeostasis. Data show that INT-777 tends to decrease the protein accumulations of ACC and P-ACC what might clearly participate to the lowering effect of INT-777 on TG concentrations.

  1. In the discussion section, the authors mentioned that “TGR5 mainly acts through the activation of the adenylate cyclase, leading to the activation of the PKA pathway and by the end to the phosphorylation of CREB. This PKA-CREB pathway has been demonstrated for decades to control testosterone production in response to the LH/CG signal”. It will be critical to demonstrate this by examining whether TGR5 and the PKA agonist H89 affect CREB phosphorylation under these conditions. There are well-established methods and materials (antibodies) for such studies.

We performed the requested experiment and analyzed the level of Phospho-CREB in response to INT-777 and H89. Data show that both molecules led to a decrease of P-CREB see figure 2 and Figure S1.

Minor comments:

  1. Please follow the guideline and nomenclature format for IJMS. For instance, gene names should be italicized in Figs 1, 2, 6, and throughout the main text. In Fig. 1A legend, PCR should be RT-PCR, and the same in the main text (Line 60). Changes have been done in the revised version.
  2. Other than the gene names in qPCR, the legend should not be italicized all through Fig. 2-6. Changes have been done in the revised version.
  3. In Line 78, should it be “3hr” instead of “1hr”? Changes have been done in the revised version.
  4. In Fig. 3B, to be consistent with the description in Line 122, it is suggested to add the “NS” between hCG and INT-77+hCG treatment.  Changes have been done in the revised version.
  5. The description of Figures 2B and 2C should be switched in lines 102 and 105. Change has been done in the revised
  6. What is the difference between the “vehicle” in Fig. 2E? Sorry in this original Fig2E, there was an error the grey bar was in fact INT-777. It has been corrected in the revised version.
  7. In line 134, “hr” should be added after the numbers 2 or 3. Change has been done in the revised 11.   In Fig. 4, there is only one panel. Thus, “in all panels” in line 157 should be deleted. C Change has been done in the revised
  8. In Line 169, “slight” should be “significant” Change has been done in the revised 13.   In Fig. 6C, “3hr” is suggested to be labeled. Change has been done in the revised version.
  9. In line 203, “use” should be changed to “us”. Change has been done in the revised 15.   In line 295, “mice” should be “Mice” starting a new sentence. Change has been done in the revised
  10. Line 307, “2” at the end of the sentence should be deleted. Change has been done in the revised
  11. In Table 1, the Tgr5 primers used should be included. Change has been done in the revised

  1. To be consistent, the title should also include “TGR5” rather than “GPBAR1”. Change has been done in the revised

Reviewer 3 Report

In this original manuscript, authors are  evaluating the implication of the G-protein-coupled bile acid receptor TGR5 in Leydig cells’ capacity to produce testosterone using the agonist INT-777. Overall, the manuscript needs proofreading for grammatical errors and lacks additional experiments to draw better conclusions on the influence of TGR5 activation on testosterone production. Thus, I have the following comments :

Major comments:

In the title, GPBAR1 should be replaced by TGR5 to be consistent with the text of the manuscript.

Line 96-97, it is mentioned « As TGR5 could modulate the protein kinase (PKA) pathway ». In which way does TGR5 modulates PKA? Please cite the reference or show the results supporting this affirmation.

Data in figure 2 on the regulation of steroidogenesis by TGR5 is non conclusive and would benefit from additional experiments. Cell lines have a reduced capacity to produce testosterone compared to normal Leydig cells. Effects of INT-777 may be more important on progesterone production, than observed on testosterone levels.

Although none of the gene expressions related to cholesterol or steroid metabolism have been altered by INT-777, changes in intracellular cholesterol and/or steroidogenesis may be attributed to changes in post-translational modifications. For example, changes in STAR phosphorylation could be investigated in response to INT-777 and would be attributed to a gene expression independent change in its activity (PMID: 10548884).

Methodology for intracellular cholesterol quantification is lacking.

Minor comments:

Line 60, replace intertitial with interstitial.

Figure 1B shows the increase in testosterone production in mLTC1 treated with hCG from 3h. However, in line 78 authors conclude that testosterone levels increase after 1h of treatment. This should be revised. 

Figure 1, the order of panels in C should correspond the the presentation in the figure legend and text. 

The image for Table 1 should be replaced by a table in word or other text format.

Author Response

Comments and Suggestions for Authors

In this original manuscript, authors are  evaluating the implication of the G-protein-coupled bile acid receptor TGR5 in Leydig cells’ capacity to produce testosterone using the agonist INT-777. Overall, the manuscript needs proofreading for grammatical errors and lacks additional experiments to draw better conclusions on the influence of TGR5 activation on testosterone production.

As requested by this reviewer, the manuscript has been completely checked for proofreading for grammatical errors.

Thus, I have the following comments :

Major comments:

In the title, GPBAR1 should be replaced by TGR5 to be consistent with the text of the manuscript. The change has been done in the title to make it consistent with the text.

Line 96-97, it is mentioned « As TGR5 could modulate the protein kinase (PKA) pathway ». In which way does TGR5 modulates PKA? Please cite the reference or show the results supporting this affirmation. We added references on that point.

Data in figure 2 on the regulation of steroidogenesis by TGR5 is non conclusive and would benefit from additional experiments. Cell lines have a reduced capacity to produce testosterone compared to normal Leydig cells. Effects of INT-777 may be more important on progesterone production, than observed on testosterone levels.

We performed new experiments and added new data confirming the decrease of testosterone levels in intracellular and medium of MLTC1 following INT-7777.

In addition, as requested we performed the measurements of progesterone measurements in INT-777 experiments. Data showed that in MLTC1 cells as well as in primary culture cells, INT-777 led to lower progesterone levels. See figure 2 and 8.

Although none of the gene expressions related to cholesterol or steroid metabolism have been altered by INT-777, changes in intracellular cholesterol and/or steroidogenesis may be attributed to changes in post-translational modifications. For example, changes in STAR phosphorylation could be investigated in response to INT-777 and would be attributed to a gene expression independent change in its activity (PMID: 10548884).

We agree with this comment, however, the antibody is not available commercially, moreover, in the time scale of this review process we cannot have access to experiment to test this hypothesis, we thus amended our comments in conclusion part, and we added a comment on the possibility that TGR5 signaling pathway and the phosphorylation of STAR.

Methodology for intracellular cholesterol quantification is lacking.

Minor comments:

Line 60, replace intertitial with interstitial. Change has been made.

Figure 1B shows the increase in testosterone production in mLTC1 treated with hCG from 3h. However, in line 78 authors conclude that testosterone levels increase after 1h of treatment. This should be revised.  Change has been made.

Figure 1, the order of panels in C should correspond the the presentation in the figure legend and text. Change has been made.

The image for Table 1 should be replaced by a table in word or other text format. Change has been made.

Round 2

Reviewer 1 Report

Authors faithfully revised the manuscript according to this reviewer’s comments and suggestions.

There are some typos.

In lines 88, 395, 422,and 432, “Nacl” should be “NaCl”.

In Table 1, “Hmgcoa-reductase” may be “Hmgco-reductase”

Reviewer 2 Report

In this revised manuscript, the authors have satisfactorily addressed the reviewer's comments.

Reviewer 3 Report

The new version of the manuscript has been improved. I have no further comments.